# A Comprehensive Identification and Expression Analysis of VQ Motif-Containing Proteins in Sugarcane (*Saccharum spontaneum* L.) under Phytohormone Treatment and Cold Stress

**DOI:** 10.3390/ijms23116334

**Published:** 2022-06-06

**Authors:** Ying Liu, Xiaolan Liu, Dandan Yang, Ze Yin, Yaolan Jiang, Hui Ling, Ning Huang, Dawei Zhang, Jinfeng Wu, Lili Liu, Liping Xu, Mingli Yan, Youxiong Que, Dinggang Zhou

**Affiliations:** 1Hunan Key Laboratory of Economic Crops Genetic Improvement and Integrated Utilization, School of Life Science, Hunan University of Science and Technology, Xiangtan 411201, China; ly1071457534@163.com (Y.L.); liuxiaolan1018@126.com (X.L.); ydd2595836963@163.com (D.Y.); yz18390246279@163.com (Z.Y.); jyl0480_17@163.com (Y.J.); zhangdawei.hnust@foxmail.com (D.Z.); wujinfeng.hnust@foxmail.com (J.W.); lilyliu@hnust.edu.cn (L.L.); ymljack@126.com (M.Y.); 2Key Laboratory of Integrated Management of the Pests and Diseases on Horticultural Crops in Hunan Province/Key Laboratory of Ecological Remediation and Safe Utilization of Heavy Metal-Polluted Soils, College of Hunan Province, Xiangtan 411201, China; 3College of Agriculture, Yulin Normal University, Yulin 537000, China; linghuich@ylu.edu.cn (H.L.); hningch@ylu.edu.cn (N.H.); 4Key Laboratory of Sugarcane Biology and Genetic Breeding, Ministry of Agriculture/National Engineering Research Center for Sugarcane, Ministry of Science and Technology, Fujian Agriculture and Forestry University, Fuzhou 350002, China; xlpmail@126.com; 5Crop Research Institute, Hunan Academy of Agricultural Sciences, Changsha 410000, China

**Keywords:** *Saccharum spontaneum*, *VQ* gene, genome-wide analysis, Transcriptome, phytohormone, cold stress

## Abstract

The VQ motif-containing proteins play a vital role in various processes such as growth, resistance to biotic and abiotic stresses and development. However, there is currently no report on the *VQ* genes in sugarcane (*Saccharum* spp.). Herein, 78 *VQ* genes in *Saccharum spontaneum* were identified and classified into nine subgroups (I-IX) by comparative genomic analyses. Each subgroup had a similar structural and conservative motif. These *VQ* genes expanded mainly through whole-genome segmental duplication. The cis-regulatory elements (CREs) of the *VQ* genes were widely involved in stress responses, phytohormone responses and physiological regulation. The RNA-seq data showed that *SsVQ* gene expression patterns in 10 different samples, including different developmental stages, revealed distinct temporal and spatial patterns. A total of 23 *SsVQ* genes were expressed in all tissues, whereas 13 *SsVQ* genes were not expressed in any tissues. Sequence Read Archive (SRA) data showed that the majority of *SsVQs* responded to cold and drought stress. In addition, quantitative real-time PCR analysis showed that the *SsVQs* were variously expressed under salicylic acid (SA), jasmonic acid (JA), abscisic acid (ABA) and cold treatment. This study conducted a full-scale analysis of the *VQ* gene family in sugarcane, which could be beneficial for the functional characterization of sugarcane *VQ* genes and provide candidate genes for molecular resistance breeding in cultivated sugarcane in the future.

## 1. Introduction

Sugarcane (*Saccharum* spp. hybrid) is the most important crop for sugar-producing and raw material for fuel alcohol around the world [1,2]. It contributes to over 80% of the sugar and approximately 40% of the ethanol global production [1,3]. The most modern sugarcane varieties have been derived from hybrids between *S. spontaneum* and *S. officinarum* [4]. *S. officinarum* (called “noble cane”) has higher sugar content, while *S. spontaneum* contributes with disease resistance genes and ratooning ability [5]. Various stresses, including pathogen infection, drought, pest attack, cold and salinity stresses, severely and consistently reduce sugar production [6,7,8,9]. Basnayake et al. reported that drought had reduced sugarcane production by up to 60% [10]. Mosaic, one of the main viral sugarcane diseases, has resulted in stunted plants, reduced photosynthesis, destruction of chlorophyll and inhibition of plant growth and which can ultimately reduce sugarcane yield by 10–50%, or even 60–80% [11,12]. Approximately 1300 insect pests have been accounted for attacking sugarcane around the world, and stem borer is the most harmful pest in most countries, resulting in yield losses of nearly 25–30% [1,13,14]. Like all plants, sugarcane has evolved fancy mechanisms to respond to external stimuli [15]. These responses formed sophisticated signaling pathways, involving many endogenous and exogenous signal molecules, such as phytohormones and other regulators, which systematically modulate transcriptional and post-transcriptional processes [15,16]. To understand the sugarcane growth and development and response to environmental stressors, it is significant to investigate the complexity of transcription factors (TFs) and transcription-associated proteins (TAPs) present in the genome and the functions [4,17].

One of the largest transcriptional regulator families in plants is the VQ proteins that work solely or in combination with other TFs to regulate the various life processes, including plant growth and development and responses to different stress [18,19,20]. The VQ proteins, named by the highly conserved amino acid motif (FxxVQxhTG), have been widely found in monocotyledons and dicotyledons [19,21,22,23,24,25,26,27]. In addition, the types of VQ proteins are varied among plants. For example, there are four types in rice (LTG, FTG, VTG and ITG) and six types in tobacco (LTG, FTG, YTG, VTG, LTA and LTV) [22,28]. Though previously thought to be plant-specific TFs, VQ proteins have also been found in bacteria, animals and fungi, implying that the *VQ* gene family has a common ancient ancestor [29].

VQ proteins play a vital regulatory role in seed development, as well as vegetative plant growth, biotic and abiotic stress responses. *IKU1* (*AtVQ14*/AT2G35230) is involved in regulating endosperm development and seed growth [30]. *AtVQ22* could alleviate JA stress through interaction with *AtWRKY28* and *AtWRKY51* [31]. Dong et al. reported that overexpression of apple *VQ37* could affect vegetative growth and reproductive growth in *Arabidopsis* and tobacco plants [27]. The *AtVQ15* plays a negative role in response to osmotic stress [32]. Meanwhile, increasingly more studies have shown that many VQ proteins interact with WRKY TFs in response to abiotic and biotic stress. *MaWRKY26* can physically interact with *MaVQ5* to control the JA biosynthetic [20]. *OsVQ7* could improve the tolerance of various stresses and development through interaction with *OsWRKY24* [33].

At present, *VQ* genes have been identified and analyzed in many plants, including both monocotyledons and dicotyledons. *Arabidopsis, Oryza sativa* and *Zea mays* were found to have 34, 39 and 61 *VQ* genes, respectively [19,25,28]. However, there is little available information about the VQ protein gene family in sugarcane. In this study, we carried out a comprehensive analysis of the *VQ* gene family based on *S. spontaneum* AP85–441 which has been sequenced and assembled. In the present study, 78 *SsVQs* in *S. spontaneum* were identified. Their classification, gene structure, conserved domains, motif composition, chromosomal location distribution, evolution and CREs were analyzed. In addition, the expression patterns for *VQ* genes in 10 different sugarcane tissues, cold and drought stress, and under hormone (ABA, SA and JA) treatment and cold stress were investigated using RNA-seq data, SRA data and real-time quantitative polymerase chain reaction (RT-qPCR), respectively. The present study aimed to promote the functional study of *VQ* genes in sugarcane, especially for the understanding of the mechanism of *VQ* genes under phytohormone and cold stress. Moreover, our results could contribute to providing key candidate genes for molecular stress resistance breeding in sugarcane.

## 2. Results

### 2.1. Identification and Sequence Analysis of SsVQ Genes

In total, we identified 78 VQ motif-containing proteins using BlastP and HMMER software and named them from *SsVQ1* to *SsVQ78* based on their physical locations on the chromosomes. The total number of *VQ* genes in sugarcane was larger than that in *Arabidopsis* (34), rice (40) and maize (61). ExPasy prediction revealed that these 78 VQ proteins have different physical and chemical properties. The amino acid lengths ranged from 77 aa (SsVQ15) to 812 aa (SsVQ30), with an average of 258 aa and most of them were less than 300 aa (Appendix A). The molecular weights ranged from 8063.42 Da (SsVQ15) to 85,904.61 Da (SsVQ30) and their isoelectric points ranged from 4.73 (SsVQ74, SsVQ75 and SsVQ76) to 11.59 (SsVQ11). To investigate the protein hydrophobicity, the GRAVY score was conducted, and the results showed that the GRAVY scores all were negative except for the SsVQ29, indicating that most SsVQ proteins are hydrophilic. Moreover, the subcellular localization prediction showed that 44 *SsVQ* genes were located in the nucleus, 15 *SsVQ* genes were located in the chloroplasts, 13 *SsVQ* genes were located in the mitochondrion, four *SsVQ* genes (*SsVQ64*, *SsVQ74*, *SsVQ75* and *SsVQ76*) were located in the cytoplasm and two *SsVQ* genes (*SsVQ37* and *SsVQ38*) were located in the endoplasmic reticulum (Appendix A).

### 2.2. Phylogenetic Analysis and Multiple Sequence Alignment

To investigate the relationships of *SsVQ* genes among sugarcane, maize, rice and *Arabidopsis*, we conducted a phylogenetic tree based on their protein sequences (Figure 1). We found that maize has a closer relationship with sugarcane than rice and *Arabidopsis*. Based on the relationship with *AtVQs* and *OsVQs* and the domains of SsVQ protein, they were divided into nine groups, named Group I–IX. For the 78 SsVQ proteins, GroupVI contains three VQ proteins; Group VII has the largest amount, with 19 VQ proteins. Group I, II, III, IV, V, VIII and IX contain 15, 8, 6, 9, 4, 6 and 8 members, respectively.

At the same time, we performed the multiple sequence alignment and found five types of VQ specificity domain (Figure 2). The results showed that all identified SsVQ proteins contained the motif FxxhVQxhTG, while the x represents any amino acid and h represents a hydrophobic amino acid. 59/78 FxxxVQxLTG, 12/78 FxxxVQxFTG, 2/78 FxxxVQxVTG (*SsVQ6* and *SsVQ18*), 2/78 FxxxVQxITG (*SsVQ21* and *SsVQ26*), 3/78 FxxxVHQxVTG (*SsVQ66*, *SsVQ67* and *SsVQ68*). Different types of VQ domains indicated that they might have different biological functions.

### 2.3. Conserved Motifs and Gene Structures of the VQ Gene Family

To predict the function of *VQ* genes, we detected the conserved motifs. The results indicated that these 78 *SsVQs* contained 10 conserved motifs and the motif length ranged from 11 aa to 50 aa (Figure 3; Appendix A). All VQ proteins contain motif 1, which was VQ domain and each *SsVQ* member contains 1–6 conserved motifs. Moreover, an unrooted phylogenetic tree was constructed based on VQ protein sequences, suggesting that the motif classification of *VQ* genes was consistent with the phylogenetic tree. We found that most groups possess three motifs, which suggested that every group might have special functions with a highly conserved amino acid residue, while seven VQ proteins only have one VQ motif (VQ1, VQ15, VQ27, VQ34, VQ45, VQ47 and VQ64). Notably, some motifs appeared in one sub-branch of *SsVQ*. For example, motif 2 only exists in Group III. Motif 8 only exists in Group V. Through the *VQ* gene structures analysis, only three of the *SsVQs* have UTR. Interestingly, 73.08% (57/78) of *SsVQ* genes are intronless genes, which indicated that many introns might be lost during the stage of *VQ* gene evolution.

### 2.4. Chromosome Distribution and Gene Duplication Analysis

To understand the position of *SsVQ* genes on the chromosome, we conducted a chromosomal location map of these *SsVQ* genes (Figure 4). *SsVQs* are located on all sugarcane chromosomes, except chromosome 7C. A large number of *SsVQs* are located on the two ends of chromosomes. Specifically, chromosome 1B and chromosome 1D all contained six genes. Segmental or tandem duplicates in many gene families are the main expanding pattern in plants [34]. To better study the evolution of *SsVQ* genes, we further predicted the gene duplication events by the MCScanX software (Figure 5). We found that 24 pairs of *VQ* genes originated from segmental duplication, and three pairs of genes (*SsVQ3*/*SsVQ4*, *SsVQ9*/*SsVQ10* and *SsVQ67*/*SsVQ68*) were involved in tandem duplication events. The Ka and Ks rates and the Ka/Ks of these *VQ* gene pairs were calculated. Results showed that 18 pairs were <1.0, two pairs were >1.0, and the other four did not have values (Appendix A). Therefore, purifying selection might be the primary pressure during the evolutionary period of *SsVQs*. We further detected the collinear relationship among *Arabidopsis*, rice, *Sorghum bicolor* and sugarcane. The results showed that sugarcane and sorghum have a closer relationship (Appendix A).

### 2.5. Identification of CREs in the Promoter Regions of SsVQ Genes

In this study, we identified putative cis-elements of 3000 bp located on *SsVQ* genes to predict the possible gene function by using PlantCARE (Appendix A). The results showed that the numerous CREs widely distributed in the promoter region of *SsVQ* genes including stress responsiveness, phytohormone responsiveness, light responsiveness and growth and development. The light, ABA and MeJA responsive elements were the most numerous in the promoter region of *SsVQs*. Among them, light responsive elements were found in all members. In total, 77/78 *SsVQs* contained ABA response elements except the *SsVQ2*, and 76/78 *SsVQs* possessed MeJA response elements except the *SsVQ2* and *SsVQ15*, while 73/78 *SsVQs* contained anaerobic induction elements (ARE). Moreover, 53 (67.95%) *SsVQs* contained drought responsive elements (MBS), 52 (66.67%) *SsVQs* contained gibberellin-responsive elements (TATC-box, P-box and GARE-motif), 43 (55.13%) *SsVQs* contained SA responsive elements, 42 (53.85%) *SsVQs* contained auxin-responsive elements (TGA-element and AuxRR-core) and 38 (48.72%) *SsVQs* contained low-temperature response elements (LTR), while 34 (43.59%) *SsVQs* contained defense and stress responsiveness (TC-rich repeats). In addition, the circadian control element, GCN4_motif (involved in endosperm expression), circadian control element and GC-motif (involved in anoxic specific inducibility) were also identified. These results indicated that most of the *SsVQ* genes might respond to various plant biotic and abiotic stresses and play a vital role in sugarcane growth and development.

### 2.6. Protein-Protein Network of the VQ Proteins

Based on the STRING database with maize as a reference, a PPI network was constructed to explore the physical and functional association (Figure 6). Some of these proteins (GRMZM2G174650_P01-GRMZM2G023921_P01, GRMZM2G174650_P01-GRMZM2G059064_P01, GRMZM2G174650_P01-GRMZM2G118172_P01) have strong interaction. These results could provide a piece of useful information for subsequent studies on the regulatory network of VQs.

### 2.7. Expression Pattern for SsVQ Genes during Sugarcane Development Based on RNA-Seq Data

Using the public RNA-seq data, we analyzed the *SsVQ* gene expression profiles in 10 different tissues to understand spatiotemporal expression patterns (Figure 7). In general, most *SsVQs* are expressed during the stage of sugarcane growth and development but with different expression. Among the *SsVQ* genes, 13 were not expressed in all tissues, while 23 were expressed in all 10 tissues (FPKM > 0). Most of these *SsVQ* genes were found expressed in more than one detected organ. *SsVQ73* and *SsVQ69* had similar expression patterns and were highly expressed at the early and mature stages of leaf and steam, while *SsVQ37* and *SsVQ38* were highly expressed at the stage of seedling–steam and seedling–leaf. In addition, *SsVQ2*, *SsVQ16*, *SsVQ33, SsVQ257* and *SsVQ66* were higher expressed in the stage of leaf development than the other nine stages, indicating that they are related to the growth and development of corresponding tissues. These results implied that *SsVQs* may play an important role in the growth and development of sugarcane, but with individual functional modes.

### 2.8. Expression Pattern for SsVQ Genes under Cold and Drought Treatments Based on SRA Data

Since cis-elements responding to drought and cold stress widely existed in the promoters of the *SsVQ* genes. We analyzed the expression patterns for *SsVQs* under drought and cold stress to further investigate the potential functions of *SsVQ* genes (Figure 8). Under drought stress, the *SsVQs* had different expression profiles. For example, six *SsVQs* (*SsVQ20*, *SsVQ24*, *SsVQ31*, *SsVQ59*, *SsVQ62* and *SsVQ65*) were not expressed. Most *SsVQ* genes had the highest expression levels after 10 days of recovery, especially *SsVQ4*, *SsVQ9*, *SsVQ33*, *SsVQ35*, *SsVQ39*, *SsVQ69* and *SsVQ73*. While *SsVQ40*, *SsVQ58*, *SsVQ60* and *SsVQ61* had the lowest expression levels after 10 days recovery. The expression level of the *SsVQ38* gene showed a continuous downward trend after drought treatment (2, 6 and 10 days drought) and after 10 days of recovery treatment. As for cold treatment, 15 *SsVQs* were not expressed. Different from the drought treatment, the expression levels of most *SsVQs* did not change significantly but remained relatively stable. Three of the *SsVQs* (*SsVQ38*, *SsVQ69* and *SsVQ73*) consistently had the highest expression level during the treatment. These results showed that *SsVQs* may play different roles in response to drought and cold stress.

### 2.9. Expression Analyses of VQ genes in Response to Hormone Treatments Using qRT-PCR

To further determine whether the *VQ* gene expression patterns were affected by hormone treatments, we examined seven *VQ* genes and performed a qRT-PCR to analyze the expression patterns under different treatments (Figure 9). SA, JA and ABA treatment could affect the expression of all seven genes. After 12 h of treatment, the up-regulation multiple for most *VQ* genes was the highest among all treatment time points compared with the control group. Different treatments had different effects on gene expression. Upon ABA treatment, four genes (*VQ8*, *VQ30*, *VQ73* and *VQ76*) were up-regulated among all time points. Two genes (*VQ27* and *VQ44*) showed a tendency towards down-regulation in all time points. Under SA stress, *VQ27* and *VQ57* were down-regulated after 1 h of treatment and had few differences after 6 h and 12 h of treatment. In the case of JA treatment, *VQ73* was rapidly up-regulated after 1 h of treatment. Interestingly, *VQ8* was dramatically up-regulated among all hormone treatments. The expression pattern for *VQ76* was the same under the three hormones treatment, which increased first and then decreased, and the expression level was the highest at 6 h. The results indicated that *SsVQs* may play an important role in response hormones.

### 2.10. SsVQ Gene Expression following Abiotic Treatments Using qRT-PCR

The above data suggested that *SsVQ* gene expression was affected by hormone treatment; we also investigated their expression levels under the cold environment (Figure 10). The expression trend for the three genes (*VQ44*, *VQ73* and *VQ76*) was the same, which increased first after 1 h of treatment and then decreased continuously after 6 h and 12 h of treatment. However, the expression of *VQ8* decreased under cold stress, which was different from that under hormone treatment. The expression pattern for *VQ30* and *VQ57* was continuously up-regulated among all treatment time points.

## 3. Discussion

VQ protein, a type of plant specific protein, is involved in plant development and can respond to different stresses [25,27,35]. Hence, we conducted a completed genome-wide analysis of sugarcane VQ proteins using bioinformatics analysis and qRT-PCR to understand their regulation with the environment changes.

In this study, 78 *SsVQ* genes were identified, which was higher than that in rice (39), maize (61) and *Arabidopsis* (34). The genome size varies in different plants, rice (466 Mb/39 *VQs*) [36], *Arabidopsis* (125 Mb/34 *VQs*) [37], maize (2.3 Gb/61 *VQs*) [38] and *S. spontaneum* (3.36 Gb/78 *VQs*) [39], indicating that the number of *VQ* gene family members may not have an absolute correlation with genome size. While the AP85–441 was haploid and produced from the octoploid SES208, the number of *VQ* genes in octoploid *S. spontaneum* could be over 78 [5]. The gene structure analysis suggested that 70.5% (55/78) of *SsVQ* genes were found to be intronless. The same phenomenon has also been found in *Glycine max* (64; 85.33%) [17], *M. truncatula* (29; 90.63%) [40] and *G. raimondii* (31; 68.85%) [24]. Comparatively, these plants indicated that most *VQ* genes have lost introns during the long evolutionary period.

Based on the results of multiple sequence alignment, we found that there are four types of VQ domains for SsVQ proteins (LTG, FTG, VTG and ITG). Previous studies have shown that there are six types of motifs in *Arabidopsis* (LTG, FTG, VTG, YTG, LTS and LTD) [25], three types in grapevine (LTG, FTG, and VTG) [41], six types in maize (LTG, FTG, VTG, ITG, ATG and LTA) [19], four types in rice (LTG, FTG, VTG and ITG) [28] and seven types in tomato (LTG, FTG, VTG, LTS, LTA, YTG and HTG) [42]. Comparing the different plants of the conserved VQ domain, we found that LTG, FTG and VTG are the three most common domain types in plants. Even though maize and *S. spontaneum* are all monocot plants, they have different VQ domain variations [19]. Therefore, the variations in conserved VQ motif may be different in various species.

The SsVQ proteins all contain the conserved VQ domain, except SsVQ66, SsVQ67 and SsVQ68, in which VH core amino acids replace the VQ core amino acids. The same phenomenon has also been found in rice and maize. The VQ core amino acids are also replaced by a VH core amino acid in rice and maize (OsVQ37, OsVQ39, ZmVQ15, ZmVQ28 and ZmVQ58) [19,28]. Interestingly, the VH core amino acids only showed in monocot plants, which may indicate the evolutionary process difference between monocot and dicots. Moreover, NtVQ54, in which isoleucine (IQ) replaces the canonical caline in the conserved domain (VQ), has demonstrated a similar phenomenon [22].

The major expansion ways of plant genome were segmental and tandem duplication events [34]. In this study, 24 gene pairs originated from segmental duplication and three gene pairs were derived from tandem duplication. Similarly, the phenomenon of low tandem and high segmental duplication proportion for gene families were also founded in *A. thaliana*, maize and Chinese cabbage [19,21,25]. In addition, gene duplication may lead to gene functional redundancy, and these duplicate genes could develop divergent patterns in gene expression for stable maintenance through subfunctionalization [43,44]. In our study, some paralogs showed different expression patterns, such as *SsVQ66*/*68*, *SsVQ24*/*30* and *SsVQ1*/*15*. Furthermore, there are numerous orthologous gene pairs in *SsVQ* genes. The substitution rates of Ka and Ks are the basis for analyzing the selection pressure in gene duplication events. The most Ka/Ks values of gene pairs were < 1, suggesting that they had mainly undergone purifying selection.

Previous research has shown that the plant *VQ* gene family members have a different expression pattern, from growth and development to response to various external stresses, especially phytohormone treatment and cold stress [26]. Expression patterns for 78 *SsVQ* genes were conducted based on their tissue expression using RNA-seq data. The results showed that there were eight genes that had a relatively high expression in ten tissues, indicating that they may be related to the growth and development of plants [16,45]. Moreover, under drought and cold stress, the *SsVQs* expression levels were conducted using SRA data (Figure 8). The results showed that most of the *SsVQs* were up-regulated or down-regulated, which was consistent with the prediction that the promoter region contained a large number of LTR response element and drought response element. In addition, to verify the function of *SsVQs* in response to stress, we selected seven *SsVQs* for qRT-PCR analysis under four different stressors (SA, JA, ABA and cold). Most *SsVQ* genes were up-regulated with phytohormone treatment, and the results are consistent with *AtVQ* genes that can respond to SA treatment [25]. The expression of *SsVQ8* was robust and induced by three phytohormones of treatment, which is similar to *BrVQ23-2* and *BrVQ23-3* [21]. Under cold stress, the expression of *SsVQ57*, *SsVQ73* and *SsVQ76* were up-regulated, which consisted of SRA data. Together, these results suggested that *SsVQ* genes participate in the network of abiotic stress responses.

## 4. Materials and Methods

### 4.1. Identification and Classification of VQ Gene Family in Sugarcane

The genomic information of sugarcane was downloaded from the *S. spontaneum* AP85–441 genome (http://www.life.illinois.edu/ming/downloads/Spontaneum genome/ accessed on 4 May 2021) [5]. The protein sequences of *VQ* genes of *A. thaliana* were downloaded from the NCBI (https://www.ncbi.nlm.nih.gov/ accessed on 4 May 2021). Firstly, the *A. thaliana* VQ proteins were used as query sequences to find SsVQ proteins of the *S. spontaneum* genome in the local BLAST (10^−5^) and the VQ conserved domain (PF05678) was used as a query to explore the sugarcane proteins databases with the values (e-value) cut-off at 0.1. Next, all of the predicted VQ proteins were submitted to SMART (http://smart.emblheidelberg.de/ accessed on 6 May 2021) and NCBI Conserved Domain Database (https://www.ncbi.nlm.nih.gov/Structure/cdd/wrpsb.cgi accessed on 10 May 2021) to confirm that they all contained the VQ motif. Lastly, physical parameters of the VQ proteins, including protein length, isoelectric point (pI) and molecular weight were calculated in ExPASy (http://www.expasy.org/tools accessed on 10 May 2021). Subcellular localization was predicted using the WoLF PSORT (https://wolfpsort.hgc.jp/ accessed on 10 May 2021).

### 4.2. Phylogenetic Analysis and Multiple Sequence Alignment of SsVQ Proteins

To analyze the evolutionary relationship of the *VQ* gene families among *Arabidopsis thaliana*, *Oryza sativa*, *Zea mays* and sugarcane, AtVQ, OsVQ, and ZmVQ proteins were downloaded from phytozomes (http://www.phytozome.org accessed on 15 May 2021). The neighbor-joining (NJ) phylogenetic tree was constructed using MEGA X software with 1000 bootstrap replications [46]. Genes were classified according to *Arabidopsis thaliana* and *Oryza sativa*. Multiple alignments of the SsVQ were conducted using Jalview software with default parameter settings based on full-length proteins [47].

### 4.3. Gene Structure, Conserved Domains and Motif Composition of SsVQ Genes

Conserved motifs of the VQ proteins were identified using MEME (v.5.3.3) (http://meme-suite.org/tools/meme/ accessed on 17 May 2021) with default settings [48]. Gene structure was investigated using GSDS 2.0 (http://gsds.cbi.pku.edu.cn/ accessed on 18 May 2021) [49]. Conserved domains of VQ proteins were identified using the NCBI CDD tool [50]. TBtools (v.1.0971) was used to integrate phylogenetic trees, conserved domains results, gene structure results and conserved motifs [51].

### 4.4. Promoter Analysis of SsVQ Genes

The 3000 bp sequences upstream of the *SsVQ* genes transcriptional start site were submitted to PlantCARE (http://bioinformatics.psb.ugent.be/webtools/plantcare/html/ accessed on 20 May 2021) to identify the putative cis-elements [52].

### 4.5. Chromosomal Distribution and Duplication Analysis of SsVQ Genes

The chromosomal distribution in the identified *SsVQ* genes was obtained and visualized using TBtools software. The gene duplication events were conducted by using MCScanX [53]. The duplication events were also detected for the *SsVQ* genes. Two proteins that have more than 40% similarity and were separated by four or fewer gene loci were identified as tandem duplication, and others were identified as segmental duplications. [54]. Non-synonymous (ka) and synonymous (ks) substitutions of each duplicated *SsVQ* genes were calculated using KaKs_Calculator 2.0 [55].

### 4.6. Prediction of the Protein-Protein Interaction Network

In total, 78 VQ proteins were submitted to STRING (version 11.5, https://cn.string-db.org/ accessed on 22 May 2021) as queries, and *Zea mays* was chosen as the reference genome for blasting with default settings [56,57]. Then, the network was constructed by the best matched homologs in *Zea mays*.

### 4.7. Expression Analysis of SsVQs Based on RNA-Seq

First, different sugarcane tissue expression data, including leaves and stems at 35 days (seedling stage), 9 months (early maturity stage) and 12 months (maturity stage), was download from *Saccharum* Genome database (http://sugarcane.zhangjisenlab.cn/sgd/html/index.html accessed on 6 June 2021). In addition, under drought and cold treatments, the public data (PRJNA590595 and PRJNA636260) were download from the SRA database (https://www.ncbi.nlm.nih.gov/sra/ accessed on 6 May 2021). Fastq [58] and hisat2 [59] tools were applied to improve the sequence quality and map sequence data on the reference genome (*S. spontaneum* AP85–441), respectively. The featurCounts in Subread package and trimmed mean of M-values (TMMs) were conducted for count read and normalization [60,61]. Finally, all the expression values (FPKMs or TMMs) were used to create heat maps and cluster analysis through TBtools [51].

### 4.8. Plant Materials, Phytohormone and Cold Treatments

The sugarcane cultivar ROC22 (*Saccharum* spp. hybrid) was provided by Key Laboratory of Sugarcane Biology and Genetic Breeding, Ministry of Agriculture, Fujian Agriculture and Forestry University (Fuzhou 350002, China). Uniform 20-day-old seedlings with 3–4 fully unfolded leaves were cultured in 1/4 Hoagland nutrient solution for one week, and then these plantlets were divided into four groups. The three groups were sprayed with ABA (100 μmol/L), SA (5 mmol/L) and MeJA (100 μmol/L), respectively (Sangon Biotech, Shanghai, China). The leaves were harvested at 0, 1, 6 and 12 h. Meanwhile, one group was treated with low temperature (4 °C). Then the leaves were harvested at 0, 1, 6 and 24 h. All samples were subject to flash freezing in liquid nitrogen immediately after collection and subsequently stored in the refrigerator at −80 °C until RNA extraction.

### 4.9. Quantitative Real-Time PCR (qRT-PCR) Analysis

Total RNA was extracted from each sample using RNAiso Plus (Takara, Dalian, China) according to the manufacturer’s instructions. The cDNA synthesis was carried out with approximately 2 µg RNA using PrimeScript RT reagent Kit with gDNA Eraser (Takara, Dalian, China). The 15 μL reaction system of qRT-PCR contains 7.5 μL SYBR Green Master Mix, 0.6 μL forward primers, 0.6 μL reverse primers, 1 μL cDNA template and 5.3 μL sterile distilled water. The qRT-PCR procedure was 95 °C for 3 min, 45 cycles of 94 °C for 15 s and 60 °C for 30 s. After that, the melting curves were analyzed. For each sample, three technical replicates were conducted to calculate the averaged Ct values. The 25S genes were used as the internal control. The 2^−ΔΔCt^ method was used to calculate the qRT-PCR data. Primers used for this study are listed in Appendix A.

## 5. Conclusions

This study comprehensively analyzed the *VQ* gene family of sugarcane. A total of 78 *SsVQs* were identified and classified in detail. Five types of VQ specificity domain were found by multiple sequence alignment, especially the FxxxVHQxVTG type, which were identified in gene *SsVQ66*, *SsVQ67* and *SsVQ68.* Gene duplication analysis showed that *VQ* genes had expanded mainly through whole-genome segmental duplication in *Saccharum spontaneum*. The promoter regions of the *SsVQ* genes contained an extremely large number of the LTR response element, drought response element and phytohormone response element, to respond to various stresses. The RNA-seq analysis showed that 23 *VQ* genes were expressed in all 10 tissues, indicating that they may be related to the growth and development of sugarcane. SRA and qRT-PCR analysis showed that genes such as *SsVQ57*, *SsVQ73* and *SsVQ76* were selected as candidate genes for agricultural purposes, which obviously respond to cold, drought and phytohormone. These results provide valuable information to understand the biological role of the *SsVQ* genes and candidate genes for molecular-assisted breeding of sugarcane.

## Figures and Tables

**Figure 1 ijms-23-06334-f001:**
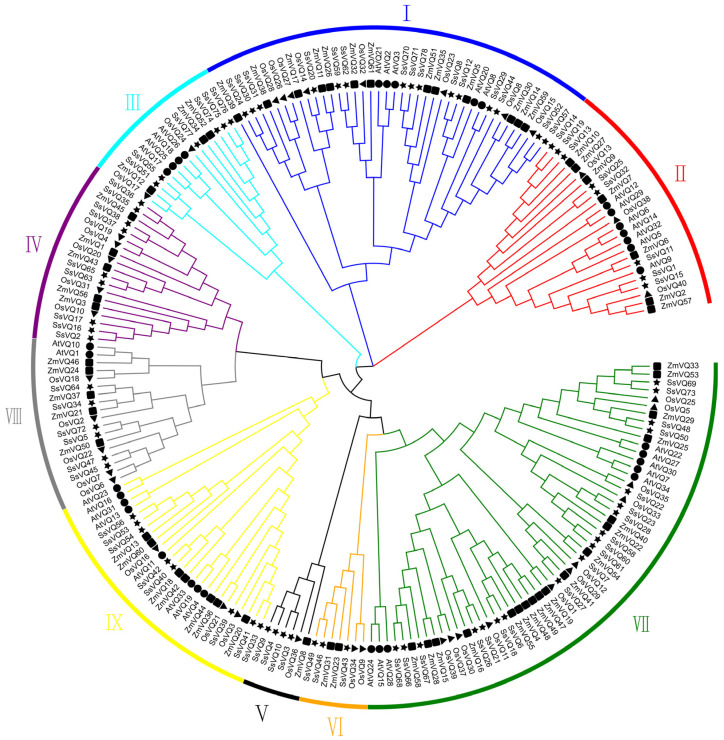
Phylogenetic analysis and family classification of the VQ domains. The NJ tree was constructed using MEGA X with 1000 boosted replicates. The different colored arcs indicate different groups of the VQ domain. Proteins from AP85-441 (*S. spontaneum*), maize (*Zea mays*), rice (*O. sativa*) and *Arabidopsis* are represented by stars, squares, triangles and circles, respectively.

**Figure 2 ijms-23-06334-f002:**
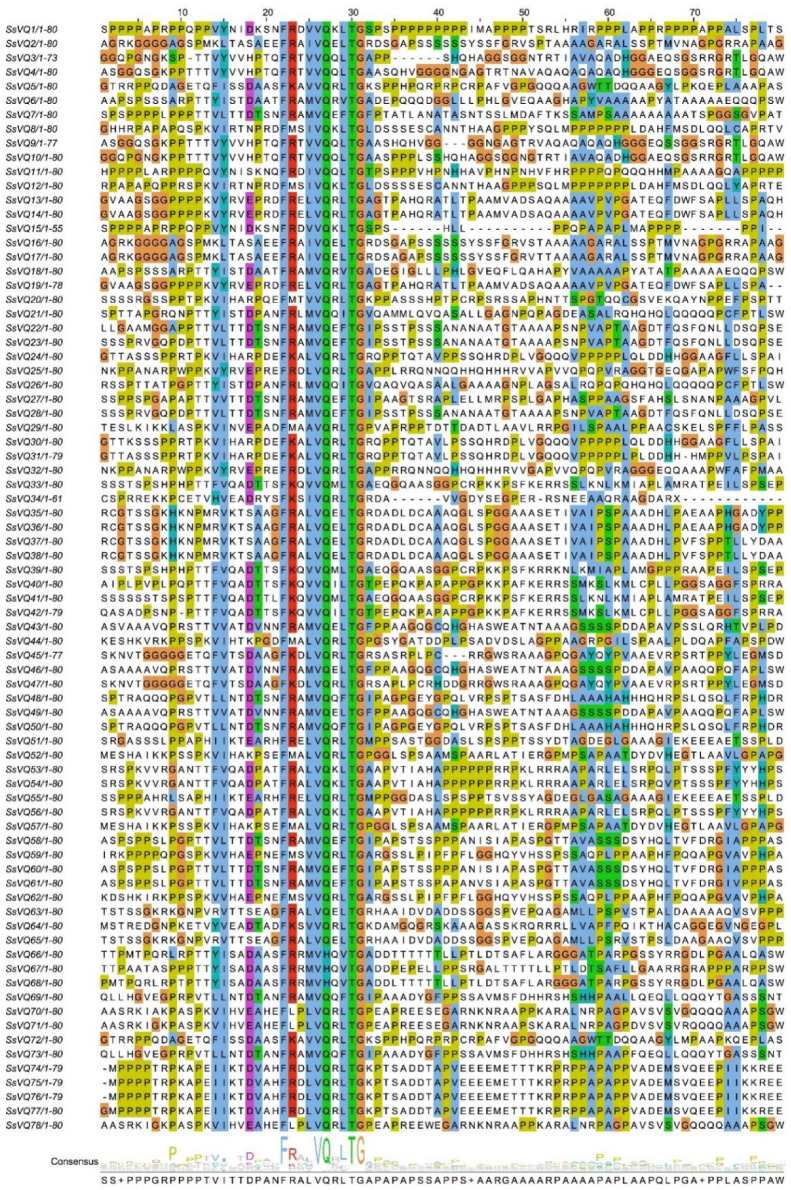
Multiple sequence alignment of the VQ proteins in *S. spontaneum***.** The sequences were aligned using the Jalview software. The highly conserved motif is FxxxVQxLTG.

**Figure 3 ijms-23-06334-f003:**
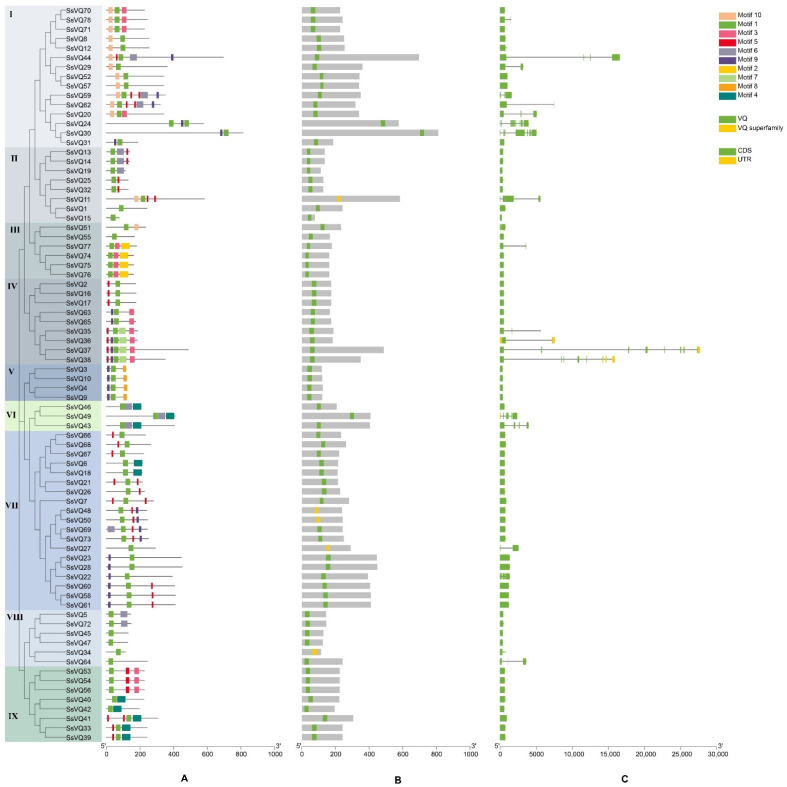
Phylogenetic tree, conserved motif, conserved domain, and the gene structure of the *VQ* genes. The phylogenetic tree was constructed using MEGA X software based on VQ protein sequences. (**A**) Conserved motifs of the VQ proteins. Each motif is represented with a specific color. (**B**) The conserved domain of VQ protein. The green and yellow boxes represent the VQ domain. (**C**) Gene structure of *VQ* gene. The untranslated 5′- and 3′-regions, introns and exons are represented with yellow box, green box and black lines, respectively.

**Figure 4 ijms-23-06334-f004:**
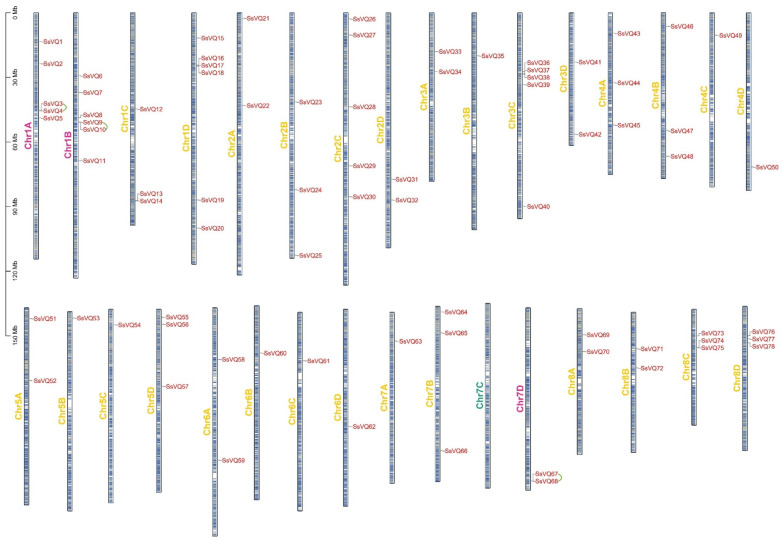
Schematic representations for the chromosomal distribution of *S. spontaneum VQ* genes. A green line between the two gene names indicated that they were tandem repeat gene pairs. The chromosome number was indicated to the left of each chromosome. *SsVQ* gene numbers are shown on the right of each chromosome. Scale bar on the left indicates the chromosome lengths (Mb).

**Figure 5 ijms-23-06334-f005:**
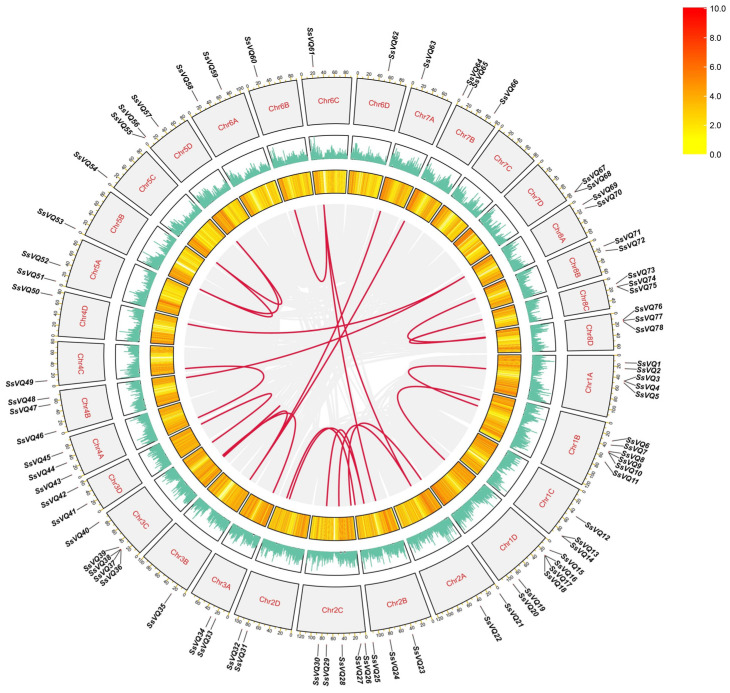
Schematic representations highlighting the interchromosomal relationships of the *SsVQ* genes. Gray lines indicated all syntenic blocks in the sugarcane genome, and the red lines indicated duplicated *VQ* gene pairs.

**Figure 6 ijms-23-06334-f006:**
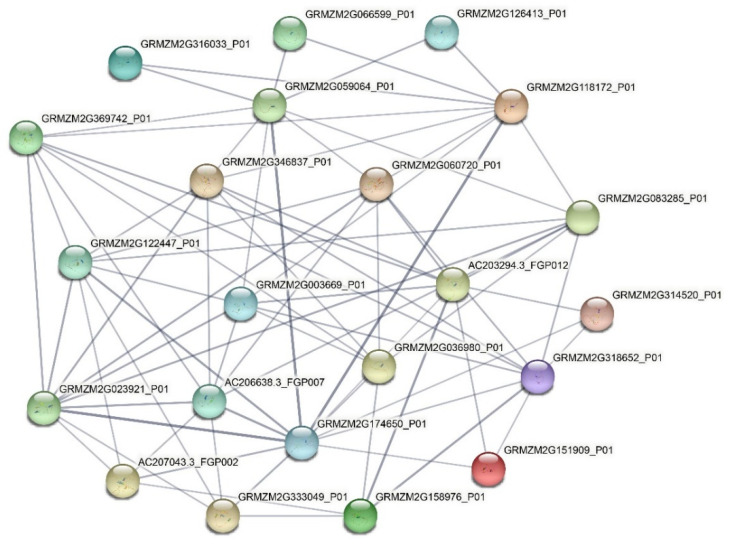
Schematic representation of protein–protein interaction (PPI) networks of the SsVQ proteins based on their orthologs in maize. The detailed information of the network is shown in Appendix A. Gray lines connect proteins within the PPI networks with darker colors and thicker lines indicating higher core PPI values.

**Figure 7 ijms-23-06334-f007:**
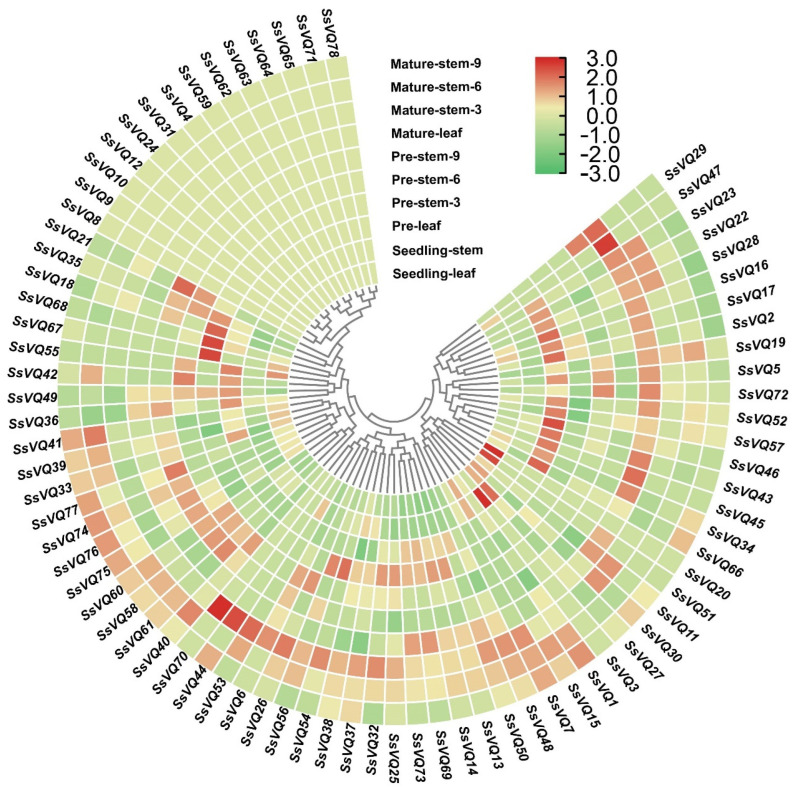
Expression profiles of the *VQ* genes from *Saccharum spontaneum* at different stages. The color bar represents the normalized values (log _2_ FPKM). The original normalized values are shown in Appendix A.

**Figure 8 ijms-23-06334-f008:**
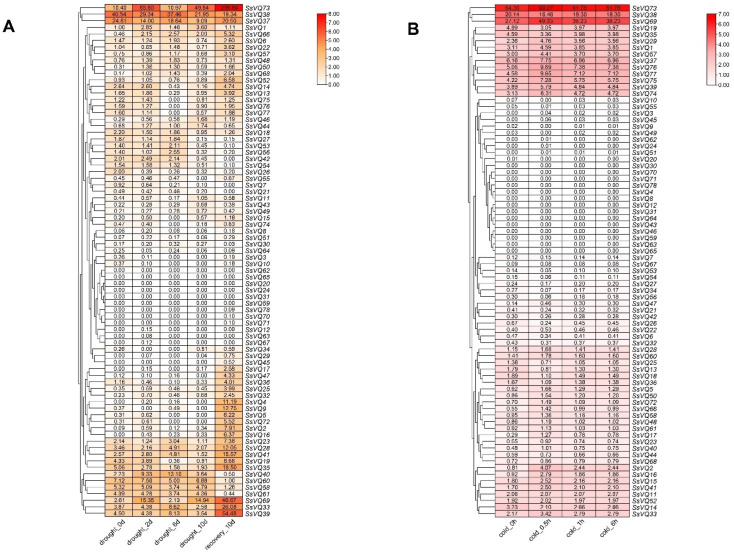
Expression profiles of *SsVQs* in response to drought and cold stresses. The number shown in the box is the original trimmed mean of M-values (TMM), which represent the expression levels of *SsVQs*. The color bar represented the normalized values (log _2_ TMM). (**A**) The heatmap of *SsVQ* genes under drought stress. (**B**) The heatmap of *SsVQ* genes under cold stress.

**Figure 9 ijms-23-06334-f009:**
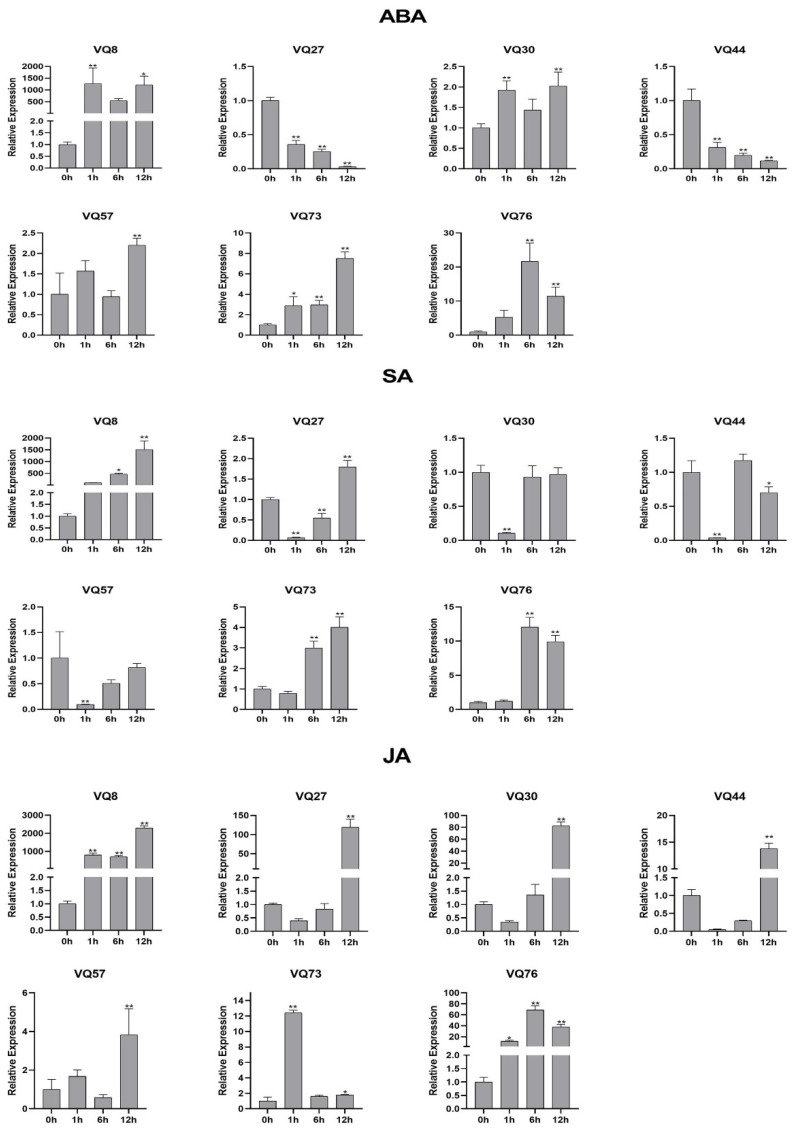
The expression profiles of seven selected *VQ* genes in response to different phytohormones. Plants were treated with ABA (100 μmol/L), SA (5 mmol/L) and MeJA (100 μmol/L) for 0, 1, 6 and 12 h. The 25s-RNA gene was used as the internal control and to normalize expression data. Relative transcript abundance was normalized relative to the expression of CK at 0 h. The 2^−ΔΔCt^ method was used to calculate the expression of target genes at different times and treatments. Error bars represented the standard deviation of the mean. ***** means *p* < 0.05, ****** means *p* < 0.01.

**Figure 10 ijms-23-06334-f010:**
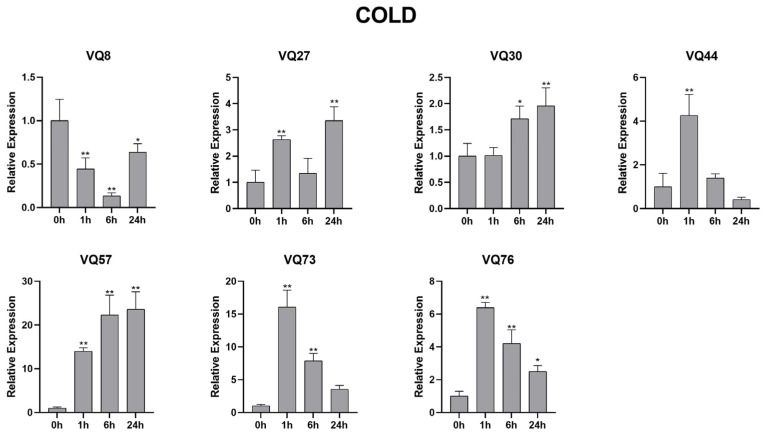
The expression profiles of seven selected *VQ* genes in response to cold stress. Plants were treated with cold (4 °C) stress for 0, 1, 6 and 24 h. The 25s-RNA gene was used as the internal control and to normalize expression data. Relative transcript abundance was normalized relative to the expression of CK at 0 h. The 2^−ΔΔCt^ method was used to calculate the expression of target genes at different times and treatments. Error bars represented the standard deviation of the mean. ***** means *p* < 0.05, ****** means *p* < 0.01.

## Data Availability

The datasets generated for this study can be found in SRA, the accession number: PRJNA590595 and PRJNA636260.

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
