# Peer review of "A Comprehensive Identification and Expression Analysis of VQ Motif-Containing Proteins in Sugarcane (Saccharum spontaneum L.) under Phytohormone Treatment and Cold Stress"

_ijms, 2022, doi:10.3390/ijms23116334_

Round 1
Reviewer 1 Report
This manuscript presents results of a thoroughly conducted project for investigating mechanisms of genes coding for tolerance of sugarcane to various stresses. The experiment was conducted using latest techniques with the target plant (not using a model such as Arabidopsis), which supports implication that this is first to show this gene family function in sugarcane. The study is very critical for not only understanding the mechanism of gene function but for potential contribution for breeding future cultivars with stress tolerance.
Author Response
Dear Reviewer:
Thank you all for taking time to read our manuscript and your affirmation of the execution and results for our manuscript. Based on suggestions from other reviewers and editors, we further revised and improved the manuscript.
Thank you again for your quick processing of the manuscript. What you have done will be greatly appreciated. Any questions, I will be more than happy to answer.
Regards,
Hunan Key Laboratory of Economic Crops Genetic Improvement and Integrated Utilization, School of Life Science, Hunan University of Science and Technology, Xiangtan 411201, China
Dr. Youxiong Que and Dinggang Zhou
2022-05-30
Reviewer 2 Report
The paper describes the family of VQ motif-containing proteins, involved in many developmental processes in plants under stress conditions. Authors have good experience in the identification of VQ genes in different plant species, like Nicotiana tabacum or Solanum lycopersicum, and they transferred their research skills and knowledge to the present study of gene expression in sugarcane (Saccharum spontaneum). Besides databases and software exploration and prediction, the work has been based on laboratory experiments, i.e. quantitative PCR. As a result, 36 VQ putative genes were selected that could be relevant for agricultural purposes. Then, the theory has to be tested in field conditions in order to prove the role of selected 36 candidate genes for molecular assisted breeding of sugarcane.
The article is original, has good technical quality, and high general interest, especially in crop production. However, some amendments are needed (highlighted in the PDF version and here within).
About the content:
The title, abstract and keywords clearly reflect paper's content. However, some improvements can be made:
L 4 – correct the editing of “spon-taneum”, should be in one word
L 29 – add “state” to “controlling physical”
L 32 – first time mentioned, explain the abbreviation “SRA” for readers
Introduction presents the problem clearly. Only:
L 84 – try to avoid starting a new phrase with a number
Results are widely described. In general, figures like no. 6 are unreadable because of the high density of information – I would propose to move Fig 6 to the supplementary materials. The schematic representation of chromosomes is given in Figure 4, and that is the most relevant for this article.
L 163 – add that the Chr 7C is not shown.
Table S7 – column “strategy” is not necessary. Put this information to the caption of the Table, i.e. “Primers used in qRT-PCR analysis”
Discussion part is justified.
Experimental methods are adequate.
L 376 – the given address of WoLF PSORT software is rather confusing. Shouldn’t be rather https://wolfpsort.hgc.jp/? Please confirm.
L 423 – mention the stage of development in days or weeks? of the tested seedlings. It is unclear, whether “groups” of seedlings or leaves were treated?
L 426 – how many leaves were subjected to the cold stress treatment? Please clarify
Conclusion is too general and it should be more focused on the obtained results:
L 442 – the style suggests that sugarcane name is VG, not gene family – reformulate
L 443-44 - more details could be given, e.g. how many genes (just in numbers, not names) could be putative candidates for a given stress factor (phytohormones miming pathogen infection, drought, and cold), priming them for future molecular assisted selection of the most resistant sugarcane plants.
References are adequate but incomplete:
There is no position no.61 from line 417.
About Presentation:
Length is commensurate with the paper's content. Quality of figures and tables is adequate.
Must be improved: conclusions not really supported by results. The manuscript is plenty of detailed data about sequence features and their expression within studied stress factors, so it is worth highlighting the most important finding at the end.
I recommend publishing the article after minor corrections.

Author Response
Dear Reviewer:
We gratefully thank you for your time spend making their constructive remarks and useful suggestions. We have carefully taken your comments into consideration in preparing this revision, which has resulted in a paper that is clearer and more compelling. Without these comments and suggestions, this manuscript would not be that smooth as what it is now. The point-by-point response to your comments and suggestions,please see the attachment.
